# Acute Liver Failure after Ingestion of Fried Rice Balls: A Case Series of *Bacillus cereus* Food Poisonings

**DOI:** 10.3390/toxins14010012

**Published:** 2021-12-23

**Authors:** Nikolaus Schreiber, Gerald Hackl, Alexander C. Reisinger, Ines Zollner-Schwetz, Kathrin Eller, Claudia Schlagenhaufen, Ariane Pietzka, Christoph Czerwenka, Timo D. Stark, Markus Kranzler, Peter Fickert, Philipp Eller, Monika Ehling-Schulz

**Affiliations:** 1Intensive Care Unit, Department of Internal Medicine, Medical University of Graz, 8036 Graz, Austria; schreibernikolaus95@gmail.com (N.S.); gerald.hackl@medunigraz.at (G.H.); alexander.reisinger@medunigraz.at (A.C.R.); 2Division of Infectious Diseases, Department of Internal Medicine, Medical University of Graz, 8036 Graz, Austria; ines.schwetz@medunigraz.at; 3Division of Nephrology, Department of Internal Medicine, Medical University of Graz, 8036 Graz, Austria; kathrin.eller@medunigraz.at; 4Department for Nutritional Microbiology, Institute for Medical Microbiology and Hygiene, Austrian Agency for Health and Food Safety (AGES), 8010 Graz, Austria; claudia.schlagenhaufen@ages.at (C.S.); ariane.pietzka@ages.at (A.P.); 5Institute for Food Safety Vienna, Austrian Agency for Health and Food Safety (AGES), 1220 Vienna, Austria; christoph.czerwenka@ages.at; 6Food Chemistry and Molecular Sensory Science, Technische Universität München, 85354 Freising, Germany; timo.stark@tum.de; 7Institute of Microbiology, University of Veterinary Medicine Vienna, 1210 Vienna, Austria; markus.kranzler@vetmeduni.ac.at (M.K.); Monika.Ehling-Schulz@vetmeduni.ac.at (M.E.-S.); 8Division of Gastroenterology and Hepatology, Department of Internal Medicine, Medical University of Graz, 8036 Graz, Austria; peter.fickert@medunigraz.at

**Keywords:** acute liver failure, fulminant hepatic failure, food poisoning, *Bacillus cereus*, cereulide

## Abstract

*Bacillus cereus* foodborne intoxications and toxicoinfections are on a rise. Usually, symptoms are self-limiting but occasionally hospitalization is necessary. Severe intoxications with the emetic *Bacillus cereus* toxin cereulide, which is notably resistant heat and acid during cooking, can cause acute liver failure and encephalopathy. We here present a case series of food poisonings in five immunocompetent adults after ingestion of fried rice balls, which were massively contaminated with *Bacillus cereus*. The patients developed a broad clinical spectrum, ranging from emesis and diarrhoea to life-threatening acute liver failure and acute tubular necrosis of the kidney in the index patient. In the left-over rice ball, we detected 8 × 10^6^
*Bacillus cereus* colony-forming units/g foodstuff, and cereulide in a concentration of 37 μg/g foodstuff, which is one of the highest cereulide toxin contaminations reported so far from foodborne outbreaks. This report emphasizes the potential biological hazard of contaminated rice meals that are not freshly prepared. It exemplifies the necessity of a multidisciplinary approach in cases of *Bacillus cereus* associated food poisonings to rapidly establish the diagnosis, to closely monitor critically ill patients, and to provide supportive measures for acute liver failure and—whenever necessary—urgent liver transplantation.

## 1. Introduction

The spore-forming facultatively anaerobic, Gram-positive bacterium *Bacillus cereus* is well known for its food poisoning potential in humans [1]. At present, there are only supportive treatments available for these food poisonings, which are attributable to various toxins. Three exotoxins, the non-haemolytic enterotoxin (Nhe), the haemolytic enterotoxin (Hbl) and the cytotoxin K (CytK), have been linked to the diarrhoeal form of food poisoning, while the small cyclic dodecadepsipeptide toxin cereulide and its isoforms are responsible for the emetic syndrome [2,3]. Among the toxins produced by *Bacillus cereus*, cereulide is the most critical, and is notably resistant to heat and acid during cooking [4]. Mild foodborne intoxications and toxicoinfections lead to emesis and diarrhoea, which generally resolve within 1–2 days. However, the emetic type of *Bacillus cereus* can occasionally also cause acute liver failure [5,6,7,8,9,10,11,12,13]. We here report a case series of food poisonings in five young and previously healthy adults after ingestion of fried rice balls that were massively contaminated with *Bacillus cereus*.

## 2. Results

The 25 years old male index patient was admitted to the intensive care unit of our university hospital with acute liver failure. His past medical history was unremarkable; he denied intake of any drugs or herbals, was fully immunocompetent, and had worked as a cook in an alpine chalet over the last six months. On the day prior to admission, he had prepared fried rice balls using rice that was boiled three days earlier and refrigerated at 8–9 °C. He tasted the food in the morning at 10:00 a.m. with a teaspoon and immediately experienced a singular episode of nausea with vomiting. He withdrew himself from the kitchen, laid down for two hours to rest, and rapidly recovered. As he did not realize the possible association of his illness with the previous food intake, he served the same rice balls as supper to himself and the whole team of the chalet at 07:00 p.m. After approximately 30 min, he and all his colleagues had emesis and developed diarrhoea. Eight hours after food ingestion, he presented to the emergency department of a local hospital. There, the initial laboratory analyses revealed leukocytosis of 18 G/L, an elevated haematocrit of 53%, and a prothrombin time INR of 2.2 (Table 1). He was immediately transferred to the intensive care unit of our university hospital because of suspected acute liver failure. Upon arrival in the tertiary care centre and 13 h after the putative food poisoning, liver transaminases were rising, the factor V was as low as 12%, and lactic acid was elevated to 9.9 mmol/L. The patient had continuous abdominal pain without cramps, while emesis and diarrhoea had already abated. The patient was initially awake and fully orientated, had a normal blood pressure of 127/57 mm Hg, a sinus tachycardia with 102 beats/minute, an oxygen saturation of 95% with ambient air, a body temperature of 37.4 °C, and a body mass index of 17.6 kg/m^2^. However, factor V decreased further to a nadir of 8% with concurrent peaks of the aspartate transaminase at 11,578 U/L, alanine transaminase at 8959 U/L, and lactate dehydrogenase at 8112 U/L 29 h after the food ingestion. Alternative causes of acute liver failure, such as acute viral hepatitis, Wilson disease and Budd–Chiari syndrome were excluded, and we strictly monitored the trends over time in mentation, prolonged prothrombin time, and factor V. The patient received rapid fluid resuscitation because of volume depletion, lactulose, *L*-ornithine-*L*-aspartate plus rifaximin (400 mg three times a day) to mitigate enteral ammonia production, a proton pump inhibitor, pre-emptive piperacillin/tazobactam therapy (4.0 g/0.5 g three times a day) because of the suspected food poisoning, and intravenous *N*-acetylcysteine to replenish mitochondrial and cytosolic glutathione stores. Ammonia and bilirubin reached the maximum value after 69 h, and 82 h, respectively (Table 1). The peak of the ammonia coincided with the lowest score in the Richmond Agitation Sedation Scale of −3 and the highest hepatic encephalopathy grade 1. In addition to the acute liver failure, the patient also developed a mild ARDS, and an acute kidney failure stage 1 with a peak serum creatinine of 1.4 mg/dL 41 h after food intake. The fractional excretion of sodium was 3%, the urinary protein/creatinine ratio (PCR) was 438 mg/g, the albumin/creatinine ratio (ACR) was 64 mg/g, and the β2- microglobulin/creatinine ratio was 82,236 μg/g. We considered orthotopic liver transplantation as a rescue strategy, but the patient did not entirely fulfil the respective Kings College or Clichy criteria. He had no flapping tremor throughout the treatment period, and rapidly recovered from acute liver failure. The patient was transferred from the ICU to a normal ward after five days and was dismissed from hospital three days later.

In parallel, four young (22 ± 2) previously healthy immunocompetent female patients were admitted to another regional hospital because of the same food poisoning event. Within 30 min after food intake, all four patients had developed emesis and subsequently diarrhoea. They collectively presented to the emergency department 8 h after the intoxication, had a normal heart rate of 71 ± 5 beats/minute, but were dehydrated. The sequential laboratory analyses revealed a white blood cell and haematocrit peak after 8 h, a slight elevation of the serum bilirubin, aspartate transaminase, and lactate dehydrogenase after 19–25 h, and a nadir of the prothrombin time after 37 h (Table 2). Ultimately, all four patients recovered with supportive treatments without ICU care and were dismissed without sequels from hospital after 2–3 days.

In the left-over rice ball, we detected a massive concentration of gram-positive rods that were subsequently identified as *Bacillus cereus* (Figure 1a,b). After 24 h of incubation, the colonies were pink (mannitol negative) with a precipitation zone (lecithinase positive) on selective MYP agar and beta-haemolytic on Columbia blood agar plates. *Bacillus cereus* was present in a concentration of 8 × 10^6^ colony forming units (cfu)/g foodstuff. All isolates tested (30 out of 30) had the identical toxin gene profile. All of them were tested positive for the cereulide non-ribosomal peptide synthase gene *ces* and the non-haemolytic enterotoxin genes *nhe* (Figure 1c,d), and all were negative for the cytotoxin K gene *cytK* and the haemolytic enterotoxin genes *hbl* (data not shown). This toxin profile, designated toxin profile E [14], is commonly found in emetic *B. cereus* strains. 

Furthermore, all isolates subjected to Fourier-transform infrared (FTIR) spectroscopy showed similar metabolic fingerprints (data not shown), indicating that the massive contamination of the foodstuff stemmed from an emetic strain. These findings were confirmed by multilocus sequence typing (MLST) of the isolated pathogen. The etiological agent of this foodborne outbreak belonged to the *Bacillus cereus* MLST-26 group, which comprises most of the emetic *B. cereus* strains, including the emetic reference strain AH187. Core genome MLST (cgMLST) revealed that the isolates from this foodborne outbreak in Styria (Austria), one from the left-over fried rice balls (21101908) and one from the kitchen sink (21102813-TP1), differ only by one allele and thus most likely present the same strain. As shown in the minimum-spanning tree (MST) in Figure 2, 21101908/21102813-TP1 cluster in closer proximity to emetic strains, which have been isolated in the context of foodborne outbreaks in different countries, than to the non-emetic *B. cereus* type strain ATCC 14579. Thus, the genetic profiles of *Bacillus cereus* strains closely correlate with clinical presentation.

Moreover, cereulide was determined via liquid chromatography-tandem mass spectrometry (LC-MS/MS) in a concentration of 37 ± 11 μg/g foodstuff. We also measured the incorporated cereulide in serum and urine of our index patient and found cereulide in a concentration of 7 ± 1 ng/mL serum and 67 ± 7 ng/mL urine 13 h after food ingestion. The recently described isocereulides A and G were detected in trace amounts. To our knowledge, this is the first record of isocereulides in human patient samples and the first accurate quantification of cereulide by stable isotope dilution analysis (SIDA)-LC-MS/MS, fostering the need for cereulide quantitation in outbreak situations, not only in foodstuffs, but also in samples from patients.

## 3. Discussion

We here report a case with acute liver failure after exposure to *Bacillus cereus* toxins that emphasizes the potential biological hazard of contaminated rice meals that are not freshly prepared and were refrigerated for several days. There are only a few reports of similar cases in adult patients published so far [5,6,7,13], including ones with a fatal outcome [6,13]. Most of the lethal courses of *Bacillus cereus* associated food poisonings have occurred in paediatric patients [7,8,9,10,11,12], with parents generally experiencing milder courses than severely sick children [5,7,8].

In line with results from a dose finding intoxication study using a porcine model [16], the symptoms in our small case series appeared to be dose related. The index patient with acute liver failure was the only one who had ingested the fried rice balls twice, whereas his four colleagues ate them only for supper and suffered from mild gastrointestinal symptoms. So far, data on exact quantitation of cereulide in food and patient samples are rare due to the lack of appropriate methods in the past. However, compared to data from the literature reporting a biological hazard of foodstuff with a cereulide concentration of 0.19–15 μg/g foodstuff [6,17,18], the cereulide amounts in the food remnants in our current case were somewhat higher. This high amount of cereulide (37 ± 11 μg/g foodstuff) might reflect the massive contamination of the food with emetic *Bacillus cereus* (8 × 10^6^ cfu/g food). The retrospective analysis of fried rice remnants contaminated with 8 × 10^6^ cfu/g food from a fatal case of food poisoning in Japan [7] revealed about 63 μg cereulide /g food [19], which is in the range of our current case and highlights the severity of the food poisoning described here. Furthermore, the amounts of cereulide determined by SIDA-LC-MS/MS in the serum sample in our index patient (7 ± 1 ng/mL serum) on admission was in the same range as the serum cereulide levels (estimated to be approx. 4 ng/mL based on Hep2 cytotoxicity assay) reported from the fatal case in Japan mentioned above. These data underpin the importance of measuring the amounts of cereulide in patient samples in parallel to the routine blood chemistry analysis as an indicator of the severity of the food poisoning. Of note, the non-haemolytical enterotoxin *nhe* was only detected by PCR. This is a limitation of this report. 

Very high aminotransferase levels and relatively low serum bilirubin levels characterized our index case. As seen in other aetiologies of hyperacute liver failure, which are characteristically due to acetaminophen-induced injury or ischemic hepatopathy [20], our patient had a rapid onset of symptoms, great severity in terms of aminotransferase levels, but also a fast recovery within one week [7,8,10]. 

Cereulide is anticipated to be the cause of acute liver failure in the context of *Bacillus cereus* associated food poisoning, mainly by interfering with mitochondrial respiration in hepatocytes [21]. It has a clear dose-response relationship in vitro when exposed to Caco-2 or HepG2 cell lines [21] as well as in vivo in a porcine model [16]. 

From the present data, we are not able to definitely conclude whether the observed acute kidney failure was prerenal, toxic, or due to rhabdomyolysis. However, the normal fractional excretion of sodium, the high PCR/ACR ratio, as well as the high urinary β2-microglobulin as a sensitive tubular injury marker, suggest a direct toxic effect leading to an acute tubular necrosis, as it similarly occurs in mushroom poisonings and in more than two-thirds of acetaminophen overdoses [22,23]. This hypothesis is fostered by the findings from a porcine intoxication study, showing that considerable amounts of cereulide accumulate in the kidneys [16]. Furthermore, it has been recently reported that long time exposure of murine liver and kidney cells to cereulide results in cytopathogenic damage [24]. Most importantly, we detected a relevant concentration of cereulide in the urine of our index patient 13 h after intoxication. This finding may serve as a rationale for providing forced diuresis and/or renal replacement therapy to accelerate renal elimination of cereulide in critically ill patients [7]. 

Last but not least, this report emphasizes the imminent necessity for a close interdisciplinary approach to these rare, but critically ill patients suffering from food poisoning and acute liver failure. Close interactions between hepatologists, intensivists, microbiologists, and transplant surgeons are pivotal to rapidly establish the diagnosis, to closely monitor critically ill patients, to provide supportive measures for acute liver failure, and/or urgent liver transplantation whenever necessary [9,25,26]. 

## 4. Materials and Methods

### 4.1. Study Population and Study Design

We describe five patients who were simultaneously admitted to three different hospitals in Styria (Austria) with symptoms of food poisoning in August 2021. All of them were personnel members of a chalet and had eaten fried rice balls on the evening of the previous day before symptom onset. The study protocol was approved by the Institutional Review Board of the Medical University of Graz, Austria (EK-Nr. 32–641 ex 19/20; 13 November 2020), complied with the Declaration of Helsinki, and written informed consent was given. 

### 4.2. Laboratory and Toxicological Analyses

Blood cell count, serum bilirubin, aspartate transaminase, alanine transaminase, lactate dehydrogenase, C-reactive protein, procalcitonin, electrolytes, ammonia, creatinine, urea, albumin, prothrombin time, factor V, and lactate were immediately measured using a Sysmex (Sysmex Austria GmbH), or Cobas (Roche Diagnostics) analyser as appropriate. For further analyses, we centrifuged peripheral blood samples at 3000 rpm for 10 min, and stored serum and urine aliquots at −20 °C. For enumeration of *Bacillus cereus* in the left-over rice ball, we used a horizontal method for the enumeration of viable presumptive *Bacillus cereus* by means of the colony-count technique at 30 °C (ÖNORM EN ISO 7932:2020). In brief, samples were commuted in peptone water and directly plated in serial dilutions on selective and differential MYP agar (Oxoid Ltd., Basingstoke, UK) and Columbia blood agar plates (bioMérieux, Marcy-l’Étoile, France) [27]. After incubation at 30 °C for 24 h, the number of typical viable colonies were counted. Thirty *Bacillus cereus* colonies from the MYP agar were randomly picked and subjected to toxin gene profiling and FTIR spectroscopy to decipher if the food was contaminated with only one strain or a mixture of *Bacillus cereus* group strains. Toxin gene profiling and metabolic fingerprinting by means of FTIR was carried out as described previously [14,15]. Following the recommendation of the ÖNORM EN ISO 7932:2020, for *ces* detection the primers EM1F/EM1R described by Ehling-Schulz et al. [28] and the primers CK1F/CK1R/CK2F/CK2R described by Guinebretiere et al. [29] were used: EM1F 5′-GACAAGAGAAATTTCTACGAGCAAGTACAAT-3′ and EM1R5′-GCAGCCTTCCAATTACTCCTTCTGCCACAGT-3′ for *ces*; CK1F 5′-CAATTCCAGGGGCAAGTGTC-3′ and CK1R 5′-CCTCGTGCATCTGTTTCATGAG-3′ for *cytK1*; CK2F 5′-CAATCCCTGGCGCTAGTGCA-3′ and CK2R 5′-GTGIAGCCTGGACGAAGTTGG-3′ for *cytK2*; the primers NA2F 5′-AAGCIG CTCTTCGIATTC-3′ and NB1R 5′-ITIGTTGAAATAAGCTGTGG-3′ were used for *nhe* [14]; and the primers HD2F 5′-GTAAATTAI GATGAICAATTTC-3′ and HA4R 5′-AGAATAGGCATTCATAGATT-3′ for *hbl* [14]. Cereulide quantitation in food and patient samples was carried out by QuEChERS-type acetonitrile extraction [30], followed by measurement using reversed phase liquid chromatography connected to tandem mass spectrometry (LC-MS/MS) according to the ISO 18465:2017. Isotope labelled cereulide served as an internal standard for the stable isotope dilution analysis (SIDA)-LC-MS/MS [31]. Furthermore, the recently described isocereulides [32] were determined using a multiparametric SIDA- LC-MS/MS method [18]. 1 mL serum as well as urine was spiked with the internal standard (5 μL) and freeze-dried. After ethanol (1 mL) extraction [30], fractionated via solid phase extraction [16], dried under a stream of nitrogen to dryness, dissolved in 100 μL methanol and analysed via LC-MS/MS [18]. For whole genome sequencing analysis, genomic DNA was isolated from overnight cultures of an isolate originating from the left overs of the rice balls and a swab taken from the kitchen sink using the MagAttract HMW DNA Kit (Qiagen, Hilden, Germany). Paired end sequencing was performed on a MiSeq platform (Illumina Inc., San Diego, CA, USA). Library preparation was carried out using Nextera XT according to the instructions of the manufacturer (Illumina Inc., San Diego, CA, USA). Raw reads were quality controlled using FastQC v0.11.7. For assembly, raw reads were de novo assembled using SPAdes version 3.15.2 (St. Petersburg State University, Center for Algorithmic Biotechnology, Saint Petersburg, Russia). MLST and core genome MLST (cgMLST) analysis (using 1984 target genes) were carried out with Ridom SeqSphere version 8.0.1. Sequences of the two isolates have been deposited in the sequence read archive (SRA) at the National Center for Biotechnology Information, U.S. National Library of Medicine (NCBI, https://www.ncbi.nlm.nih.gov/sra, accessed on 15 October 2021) under PRJNA771514 (isolate from the rice balls: SAMN22314503, isolate from the kitchen sink: SAMN22314505).

### 4.3. Statistical Analyses

Statistical analyses were performed with SPSS 26 (SPSS Inc., Chicago, IL, USA). Categorical variables were given as absolute frequencies (%), continuous variables were summarized as means ± SEM or medians [25th–75th percentile].

## Figures and Tables

**Figure 1 toxins-14-00012-f001:**
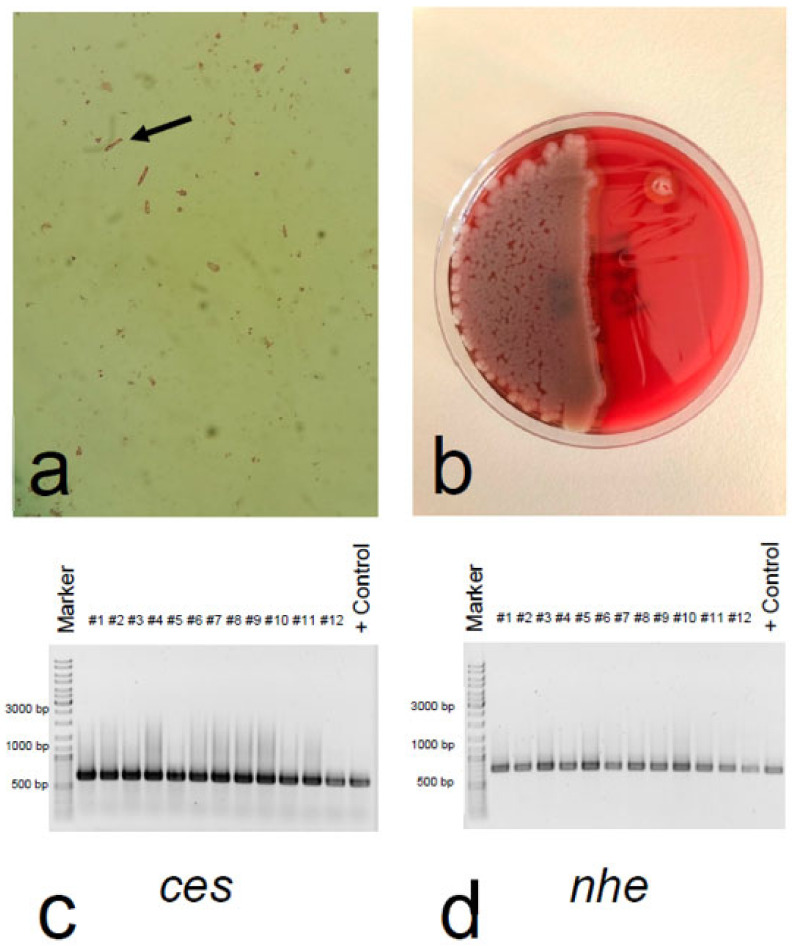
Detection of *Bacillus cereus* and toxin gene profiling (**a**) After dilution of the fried rice ball in NaCl 0.9%, Gram-staining showed Gram-positive rod-shaped bacilli with square ends (arrow). (**b**) Cultivation of the fried rice balls on blood agar plates yielded large, dull, grey, spreading colonies of *Bacillus cereus*. (**c**) Confirmation of cereulide non-ribosomal peptide synthase gene *ces*, and (**d**) non-haemolytic enterotoxin gene *nhe* by PCR in all *Bacillus cereus* isolates originating from the fried rice balls. A selection of isolates, designated #1–12, is shown together with the emetic reference strain F4810/72, which served as positive control (+Control). The expected PCR products were 635 bp for the *ces*, and 766 bp for the *nhe* gene.

**Figure 2 toxins-14-00012-f002:**
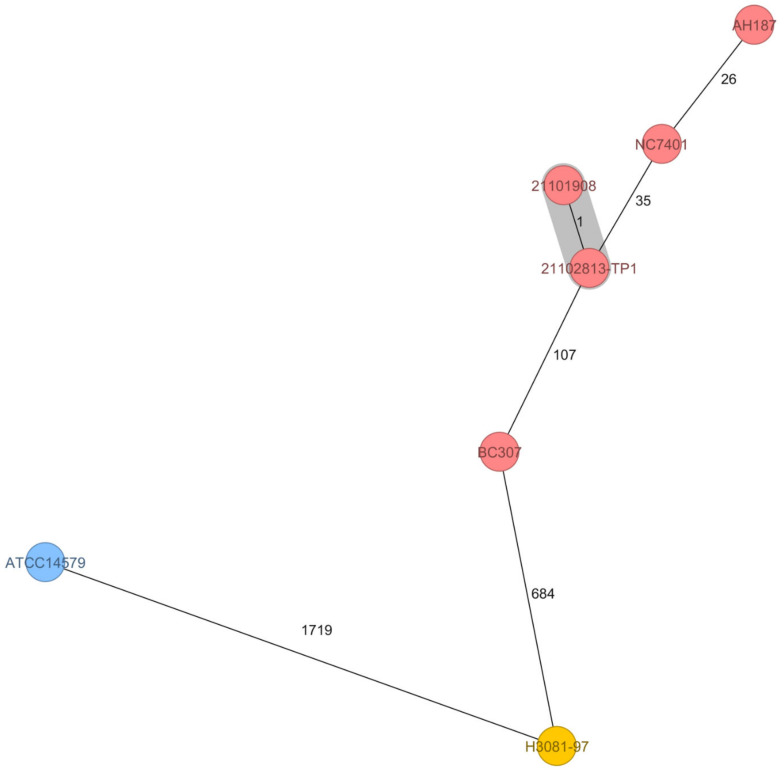
Minimum-spanning tree based on core genome multilocus sequence typing (cgMLST) allelic profiles of seven isolates of *Bacillus cereus*. The distance calculation was performed using 1984 cgMLST targets and is shown in allelic differences. The numbers on the connecting lines illustrate the number of target genes with differing alleles. Closely related genotypes (≤10 alleles difference) are shaded in grey. MLST ST-26 are given as red circles, MLST ST-4 as blue circles, and MLST ST-144 as yellow circles. The two isolates originating from the 2021 outbreak in Austria, 21101908 (from the food remnant of the fried rice balls; SAMN22314503) and 21102813-TP1 (swab from kitchen sink; SAMN22314505), differed in only one allele and cluster in proximity to strains originating from emetic outbreaks worldwide. Emetic strains originating from food poisoning included for comparison: the emetic reference strain AH187 (also designated F4810/72 [15]) isolated 1972 from vomit of a patient in UK (outbreak linked to cooked rice; SAMN02604058); NC7401 isolated 1994 from vomit of a patient in Japan SAMD00060916; BC307 isolated 2015 from vomit of a patient with fulminant liver failure in China SAMN13620036; H3081.97 isolated 1997 from food linked to a foodborne outbreak in the US ASM17103v2. The non-emetic *B. cereus* type strain ATCC 14579 served as outgroup ASM609429v1.

**Table 1 toxins-14-00012-t001:** Laboratory parameters in the index patient with severe *Bacillus cereus* intoxication.

Time after Ingestion	8 h	13 h	22 h	29 h	36 h	41 h	58 h	82 h	Normal Range
Leucocytes [G/L]	18.3	19.4	16.3	15.5	13.4	12.6	9.1	5.1	3.9–10.4
Erythrocytes [T/L]	6.1	5.5	5.0	4.9	4.7	5.0	5.0	4.1	4.0–5.2
Haemoglobin [g/dL]	18.6	16.5	15.0	14.8	14.3	15.2	14.9	12.6	11.6–15.5
Haematocrit [%]	53	48	43	43	42	44	44	37	35–45
Platelets [G/L]	310	321	274	249	219	205	174	111	140–440
CRP [mg/L]	5	9	47	65	75	86	75	54	<5.0
Procalcitonin [ng/mL]	n.d.	1.4	2.2	2.2	2.0	1.8	1.5	1.2	<0.5
Sodium [mmol/L]	143	142	136	136	137	137	141	140	135–145
Potassium [mmol/L]	4.5	4.6	4.7	4.6	4.4	4.2	3.8	4.0	3.5–5.0
Chloride [mmol/L]	103	101	99	99	99	97	99	101	98–107
Calcium [mmol/L]	2.3	2.3	2.1	2.2	2.1	2.2	2.2	2.2	2.2–2.7
Creatinine [mg/dL]	1.3	1.2	1.1	1.2	1.3	1.4	1.3	1.3	0.5–0.9
Urea [mg/dL]	38	38	35	33	33	36	45	56	10–39
Bilirubin [mg/dL]	1.1	1.5	1.8	2.3	2.7	3.1	3.2	3.6	0.1–1.2
AP [U/L]	90	95	93	95	99	105	106	76	35–105
GGT [U/L]	21	29	26	32	41	52	62	56	<40
CHE [U/L]	13,615	7416	5900	5527	5249	5410	5027	3637	>4600
AST [U/L]	61	540	9014	11,578	9973	8494	3660	1081	<35
ALT [U/L]	59	542	7856	8959	7785	7462	5361	2763	<35
Ammonia [μmol/L]	n.d.	n.d.	119	116	104	104	129	132	<50
CK [U/L]	161	163	526	1180	1799	2186	1951	1444	<170
LDH [U/L]	311	619	7453	8112	6239	5063	2328	711	<250
Prothrombin time [%]	34	30	19	17	16	17	22	38	70–130
PT INR	2.2	2.4	3.7	4.1	4.3	4.1	3.2	2.0	
aPTT [sec]	41	39	52	55	56	56	52	48	27–35
Fibrinogen [mg/dL]	191	214	153	138	136	128	124	139	180–400
Antithrombin III [%]	92	78	60	58	53	47	42	40	80–13
Factor V [%]	n.d.	12	9	8	8	9	15	35	>70
Albumin [g/dL]	4.5	4.5	3.6	3.5	3.3	3.3	3.3	3.6	3.5–5.3
Lactate [mmol/L]	n.d.	9.9	5.0	5.4	5.8	5.8	2.7	1.9	0.5–2.2

Abbreviations: CRP = C-reactive protein, AP = alkaline phosphatase, GGT = gamma-glutamyl transferase, CHE = cholinesterase, AST = aspartate transaminase, ALT = alanine transaminase, CK = creatinine kinase, LDH = lactate dehydrogenase, PT = prothrombin time, aPTT = activated partial thromboplastin time, n.d. = not determined. Abnormal parameters outside the reference range are marked in red.

**Table 2 toxins-14-00012-t002:** Laboratory parameters in four patients with mild *Bacillus cereus* intoxication.

Time after Ingestion	8 h (*n* = 4)	19 h (*n* = 4)	25 h (*n* = 4)	37 h (*n* = 4)	61 h (*n* = 3)	Normal Range
Leucocytes [G/L]	21.9 ± 1.8	13.5 ± 0.9	10.6 ± 0.5	7.2 ± 0.7	4.7 ± 0.1	3.9–10.4
Erythrocytes [T/L]	4.7 ± 0.2	4.4 ± 0.2	4.1 ± 0.1	4.0 ± 0.1	4.1 ± 0.1	4.0–5.2
Haemoglobin [g/dL]	14.0 ± 0.4	13.3 ± 0.2	12.6 ± 0.2	12.3 ± 0.3	12.1 ± 0.1	11.6–15.5
Haematocrit [%]	41.6 ± 1.1	39.0 ± 0.5	36.7 ± 0.6	36.2 ± 0.8	36.0 ± 0.2	35–45
Platelets [G/L]	302 ± 50	294 ± 44	253 ± 37	228 ± 36	226 ± 29	140–440
CRP [mg/L]	1.7 ± 1.1	15.7 ± 2.3	25.8 ± 4.4	47.5 ± 12.7	30.2 ± 8.5	<5.0
Sodium [mmol/L]	137 ± 1	136 ± 1	137 ± 1	137 ± 1	140 ± 1	135–145
Potassium [mmol/L]	3.7 ± 0.2	3.8 ± 0.1	3.7 ± 0.1	3.8 ± 0.1	4.0 ± 0.1	3.5–5.0
Chloride [mmol/L]	103 ± 1	103 ± 1	104 ± 1	103 ± 1	106 ± 1	98–107
Calcium [mmol/L]	2.2 ± 0.1	2.1 ± 0.1	2.1 ± 0.1	2.1 ± 0.1	2.1 ± 0.1	2.2–2.7
Creatinine [mg/dL]	0.8 ± 0.1	1.1 ± 0.3	0.8 ± 0.1	0.8 ± 0.1	0.7 ± 0.1	0.5–0.9
Urea [mg/dL]	35 ± 4	41 ± 7	30 ± 3	27 ± 3	18 ± 3	10–39
Bilirubin [mg/dL]	n.d.	0.7 ± 0.1	1.1 ± 0.3	0.9 ± 0.4	0.4 ± 0.1	0.1–1.2
AP [U/L]	68 ± 2	59 ± 2	65 ± 1	59 ± 3	55 ± 5	35–105
GGT [U/L]	18 ± 1	15 ± 1	15 ± 1	14 ± 1	16 ± 1	<40
AST [U/L]	37 ± 5	43 ± 4	39 ± 5	39 ± 4	35 ± 5	<35
ALT [U/L]	38 ± 1	22 ± 7	22 ± 6	21 ± 5	17 ± 3	<35
LDH [U/L]	302 ± 50	304 ± 22	248 ± 9	249 ± 16	224 ± 22	<250
Prothrombin time [%]	114 ± 7	97 ± 3	89 ± 1	87 ± 3	97 ± 4	70–130
PT INR	0.93 ± 0.03	1.01 ± 0.01	1.06 ± 0.01	1.07 ± 0.02	1.01 ± 0.03	
aPTT [sec]	n.d.	27 ± 2	27 ± 2	29 ± 2	n.d.	27–35
Fibrinogen [mg/dL]	n.d.	237 ± 7	235 ± 6	306 ± 24	n.d.	180–400
Antithrombin III [%]	n.d.	101 ± 3	96 ± 3	96 ± 2	n.d.	80–130

Abbreviations: CRP = C-reactive protein, AP = alkaline phosphatase, GGT = gamma-glutamyl transferase, AST = aspartate transaminase, ALT = alanine transaminase, CK = creatinine kinase, LDH = lactate dehydrogenase, PT = prothrombin time, aPTT = activated partial thromboplastin time, n.d. = not determined. Abnormal parameters outside the reference range are marked in red.

## Data Availability

The data presented in this study are available on request from the corresponding author. Genome sequences have been deposited in the sequence read archive (SRA) at the National Center for Biotechnology Information, U.S. National Library of Medicine (NCBI, https://www.ncbi.nlm.nih.gov/sra, accessed on 15 October 2021) under PRJNA771514 (SAMN22314503, SAMN22314505).

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
