# Peer review of "Acute Liver Failure after Ingestion of Fried Rice Balls: A Case Series of Bacillus cereus Food Poisonings"

_toxins, 2021, doi:10.3390/toxins14010012_

Round 1
Reviewer 1 Report
The authors detailly described five cases of Bacillus cereus food poisonings, including the epidemiological investigation and clinical information. From the article, patients were immediately poisoned after contaminated food ingestion with the symptoms of emesis and diarrhea. After 8~13 hours, serum bilirubin, aspartate transaminase, lactic acid, factor V and lactate dehydrogenase reached their concurrent peaks, indicating that Bacillus cereus poisoning develops rapidly. Further experiments of PCR and LC-MS/MS detection demonstrated cereulide was the cause of poisoning. The results showed a high level of biological hazard and cereulide in foodstuff compared to data from literature.
- The abstract emphasizes the food poisonings occurred in five immunocompetent adults, but it seems no mention in the subsequent contents.
- What is inherent biological hazard? We think the biological hazard of B. cereus was from air contamination or other pollution.
- We suggested that abnormal parameters in the tables should be marked, for example, ↑ represents the parameters above the normal levels and ↓ represents the parameters below the normal levels.
- What is the conclusion of minimum-spanning tree of B. cereus from different countries?
- Non-hemolytic enterotoxin was mentioned in introduction and the gene nhe was detected by PCR, however there was no verification about the generation of Nhe and no discussion at the end of the article.
Author Response
General Comment:
“The authors detailly described five cases of Bacillus cereus food poisonings, including the epidemiological investigation and clinical information. From the article, patients were immediately poisoned after contaminated food ingestion with the symptoms of emesis and diarrhea. After 8~13 hours, serum bilirubin, aspartate transaminase, lactic acid, factor V and lactate dehydrogenase reached their concurrent peaks, indicating that Bacillus cereus poisoning develops rapidly. Further experiments of PCR and LC-MS/MS detection demonstrated cereulide was the cause of poisoning. The results showed a high level of biological hazard and cereulide in foodstuff compared to data from literature.”
General Answer:
We would like to thank Reviewer 1 for his/her valuable and detailed comments.
Comment 1:
“The abstract emphasizes the food poisonings occurred in five immunocompetent adults, but it seems no mention in the subsequent contents.”
Answer 1:
We have now emphasized also in the main manuscript that all five patients were fully immunocompetent and did not take any immunosuppressive medication.
Comment 2:
“What is inherent biological hazard? We think the biological hazard of B. cereus was from air contamination or other pollution.”
Answer 2:
We absolutely agree with Reviewer 1 that rice meals do not have an inherent biological hazard. We have corrected the ambiguous statement in the abstract and the discussion of the manuscript. We now refer to the potential hazard of contaminated rice meals.
Comment 3:
“We suggested that abnormal parameters in the tables should be marked, for example, ↑ represents the parameters above the normal levels and ↓ represents the parameters below the normal levels.”
Answer 3:
We have now highlighted abnormal parameters in table 1 and 2 with arrows, as suggested by the Reviewer.
Comment 4:
“What is the conclusion of minimum-spanning tree of B. cereus from different countries?”
Answer 4:
Thank you for this comment. We now put more emphasis on the fact that the genetic sequence of Bacillus cereus and the clinical presentation closely correlate with each other. In our patients, genetic profile and clinical presentation are very similar to other reports of foodborne outbreaks with emetic strains of Bacillus cereus. Non-emetic strains such as ATCC 14579 are genetically more different and typically present with diarrhea and without emesis.
Comment 5:
“Non-hemolytic enterotoxin was mentioned in introduction and the gene nhe was detected by PCR, however there was no verification about the generation of Nhe and no discussion at the end of the article.”
Answer 5:
We absolutely agree with Reviewer 1 that we only detected the non-hemolytic enterotoxin gene nhe by PCR, and not by other methods. We now refer to this limitation in the discussion of the manuscript.
We again would like to thank the Reviewer 1 for his/her valuable feedback.
Reviewer 2 Report
Comments and Suggestions for Authors
The manuscript titled “Acute liver failure after ingestion of fried rice balls: A case series of Bacillus cereus food poisonings” lays out a very interesting description between severe intoxications with the emetic Bacillus cereus toxin cereulide on five immunocompetent adults after ingestion of fried rice balls. Characterization of Bacillus cereus foodborne intoxications is necessary to understand and rapidly establish the clinical diagnosis to improve the management of infections and intoxications caused by this microorganism. The work is well performed and easy to follow. I think that this adds to the body of knowledge and recommend for acceptance with minor comments addressed below:
In results, lines 46-75, description of some biological marker during the symptoms will be helpful to understand it. Also, preemptive piperacillin/tazobactam therapy, what is the concentration and dosis of treatment?
Figure 1, selection of isolates, designated #1-12 on the top of gel is incomplete.
Author Response
General Comment:
“The manuscript titled “Acute liver failure after ingestion of fried rice balls: A case series of Bacillus cereus food poisonings” lays out a very interesting description between severe intoxications with the emetic Bacillus cereus toxin cereulide on five immunocompetent adults after ingestion of fried rice balls. Characterization of Bacillus cereus foodborne intoxications is necessary to understand and rapidly establish the clinical diagnosis to improve the management of infections and intoxications caused by this microorganism. The work is well performed and easy to follow. I think that this adds to the body of knowledge and recommend for acceptance with minor comments addressed below.”
General Answer:
We are grateful for this detailed feedback on our manuscript and are very thankful that Reviewer 2 took the time to review our paper and for providing us with his/her valuable comments. We answered each remark in a point-by-point fashion below and corrected the manuscript accordingly.
Comment 1:
“In results, lines 46-75, description of some biological marker during the symptoms will be helpful to understand it. Also, preemptive piperacillin/tazobactam therapy, what is the concentration and dosis of treatment?”
Answer 1:
We now report additional biological markers (oxygen saturation, body temperature, and body mass index) for the index patient. Moreover, we also describe concentration and dosis of the antibiotic treatments with piperacillin/tazobactam and rifaximin in the revised version of the manuscript.
Comment 2:
“Figure 1, selection of isolates, designated #1-12 on the top of gel is incomplete.“
Answer 2:
We have corrected Figure 1 and provide the complete gel in the subsets 1c and 1d.
We once again thank you for your comments and most of all for your time to provide us with this valuable feedback that helped us to improve the manuscript.
Reviewer 3 Report
Congratulations, this study is very well done and presents -as far as I know - for the first time a broad range of laboratory parameters over a time period of up to 82 h observed in a patient after cereulide intoxication. In addition, the authors put much effort in characterization of the involved B. cereus strain and measuring toxin levels in food, serum and urine. I think these data will be of high interest to all scientist working in this field.
Author Response
General Comment:
“Congratulations, this study is very well done and presents -as far as I know - for the first time a broad range of laboratory parameters over a time period of up to 82 h observed in a patient after cereulide intoxication. In addition, the authors put much effort in characterization of the involved B. cereus strain and measuring toxin levels in food, serum and urine. I think these data will be of high interest to all scientist working in this field.”
General Answer:
We are grateful for this encouraging feedback on our manuscript and are very thankful that Reviewer 3 took his/her time to review our paper.
Reviewer 4 Report
The article “Acute liver failure after ingestion of fried rice balls: A case series of Bacillus cereus food poisonings” describes a clinical case related to food poisoning caused by the contamination of boiled rice with Bacillus cereus. It can be seen from the description that the clinical signs of acute liver failure were growing very quickly and for the most affected patient there was even a question of emergency liver transplantation. However, adequate therapy made it possible to quickly remove patients from a critical condition and everyone was safely discharged from the hospital. I am not a medical practitioner, however, in my opinion, the clinical studies and the applied therapeutic measures look quite logical and reasonable. And in general, the work was carried out at a high methodological level. Nevertheless, I have a few questions and comments for the authors.
- It is necessary to correct figure 1. Based on the caption to the picture, in the photos c. and d. Samples # 1-12 should be submitted, but in fact, the photo is cropped at Sample # 10. Also, when illustrating the results of PCR analysis, it is necessary to set a negative control in addition to the positive control. It is also advisable to move the molecular weight indicator so that the inscription is at the level of the corresponding marker. Now, according to the figure, it seems that the amplification product corresponds to 1000 bp. Signed ces under the photo c. you can remove or add the caption nhe ​​to the photo d.
- In the Materials and Methods section (lines 295-296), I was confused by the description of primers "the primers NA2F 5´-AAGCIG CTCTTCGIATTC-3´ and NB1R 5´-ITIGTTGAAATAAGCTGTGG-3´ were used for nhe [27];" As far as I know, there are several options for nhe: A, B, C. Primer names NA2F and NB1R look like “For.” taken for the nheA gene, and "Rev" for the nheB gene. The link indicated by the authors [27] contains an article (Ehling-Schulz, M .; Svensson, B .; Guinebretiere, MH; Lindbäck, T .; Andersson, M.; Schulz, A .; Fricker, M .; Christiansson, A. ; Granum, PE; Märtlbauer, E .; Nguyen-The, C .; Salkinoja-Salonen, M .; Scherer, S. Emetic toxin formation of Bacillus cereus is restricted to a single evolutionary lineage of closely related strains. Microbiology 2005, 151, 183-197.doi.org/10.1099/mic.0.27607-0), which lists separately primer pairs for nheA (NA F GTTAGGATCACAATCACCGC, NA R ACGAATGTAATTTGAGTCGC) and nheB (NB F TTAGTAGTGGATCTGTACGC NB R TAATGTTCGTTAATCCTGC). Moreover, none of the primers matches those given in the article. There is probably confusion, perhaps the primers were taken from another source. I ask the authors to clarify this discrepancy.
- Are patients being monitored after their dismissing after such intoxication? There is evidence that acute liver failure, accompanied by massive death of hepatocytes and release of “damage markers”, can lead to systemic inflammation. In this regard, for determining the outcome of acute liver failure, it is important to take into account possible immune dysfunctions, for example, to monitor the level of circulating pro-inflammatory and anti-inflammatory cytokines. If such studies are carried out, then it would be like to be mentioned in the discussion of the results.
